# Charting the Path: Navigating Embryonic Development to Potentially Safeguard against Congenital Heart Defects

**DOI:** 10.3390/jpm13081263

**Published:** 2023-08-15

**Authors:** José Bragança, Rute Pinto, Bárbara Silva, Nuno Marques, Helena S. Leitão, Mónica T. Fernandes

**Affiliations:** 1Algarve Biomedical Center-Research Institute (ABC-RI), University of Algarve Campus Gambelas, 8005-139 Faro, Portugal; 2Algarve Biomedical Center (ABC), University of Algarve Campus Gambelas, 8005-139 Faro, Portugal; 3Faculty of Medicine and Biomedical Sciences (FMCB), University of Algarve Campus Gambelas, 8005-139 Faro, Portugal; 4Champalimaud Research Program, Champalimaud Centre for the Unknown, 1400-038 Lisbon, Portugal; 5PhD Program in Biomedical Sciences, Faculty of Medicine and Biomedical Sciences, Universidade do Algarve, 8005-139 Faro, Portugal; 6School of Health, University of Algarve Campus Gambelas, 8005-139 Faro, Portugal

**Keywords:** stem cells, placenta and heart development, congenital heart diseases, secretomes, exosome, blastocyst, cardiogenic signaling pathways

## Abstract

Congenital heart diseases (CHDs) are structural or functional defects present at birth due to improper heart development. Current therapeutic approaches to treating severe CHDs are primarily palliative surgical interventions during the peri- or prenatal stages, when the heart has fully developed from faulty embryogenesis. However, earlier interventions during embryonic development have the potential for better outcomes, as demonstrated by fetal cardiac interventions performed in utero, which have shown improved neonatal and prenatal survival rates, as well as reduced lifelong morbidity. Extensive research on heart development has identified key steps, cellular players, and the intricate network of signaling pathways and transcription factors governing cardiogenesis. Additionally, some reports have indicated that certain adverse genetic and environmental conditions leading to heart malformations and embryonic death may be amendable through the activation of alternative mechanisms. This review first highlights key molecular and cellular processes involved in heart development. Subsequently, it explores the potential for future therapeutic strategies, targeting early embryonic stages, to prevent CHDs, through the delivery of biomolecules or exosomes to compensate for faulty cardiogenic mechanisms. Implementing such non-surgical interventions during early gestation may offer a prophylactic approach toward reducing the occurrence and severity of CHDs.

## 1. Introduction

The development of the human heart is a complex process that is initiated early during embryogenesis, orchestrated by intricate molecular and cellular events. Congenital heart diseases (CHDs), which result from improper embryonic development, are a group of structural and functional cardiac defects that affect approximately 1% of live births [1]. CHDs may present variable degrees of severity, ranging from minor defects with minimal or no clinical impact to severe malformations requiring immediate medical intervention at birth. Some cardiac conditions may necessitate lifelong treatment and care, frequently resulting in long-term morbidity and increased mortality rates [2]. At present, therapeutic approaches to the treatment of CHDs primarily focus on palliative interventions that improve cardiac structure and function, alleviate symptoms, and reduce the risk of long-term complications. Even surgical interventions, such as cardiac repair or transplantation, which significantly improve survival rates for patients, do not address the underlying genetic abnormalities associated with CHDs. As a result, the risk of CHDs in the children of patients who survived critical CHDs increases to 2–5% [1]. Consequently, there has been a shift from high infant mortality rates to increased survival of adults with complex CHDs, who are at risk for subsequent cardiovascular complications potentially leading to heart failure and the need for lifelong medical care [3]. In the United States, although pediatric admissions for patients with CHDs only account for 3.7% of total admissions, their annual cost is approximately $5.6 billion, which is about 15% of overall costs for pediatric patients [4]. Thus, CHDs are major and increasing health and economic burdens worldwide, and novel strategies to reliably diagnose, refine treatments for, and, if possible, prevent CHDs are urgently needed.

Current prenatal screening programs, genetic counselling, and optimization of maternal health and care, both before and during pregnancy, play a pivotal role in preventing CHDs [5]. The accuracy of prenatal tests to identify cardiac fetal anomalies and the development of innovative medical procedures have opened possibilities for fetal cardiac interventions (FCIs) in utero, at a critical developmental stage when the heart is still growing and developing. FCIs aim to correct fetal cardiac malformations that are either prone to progress with severe complications during mid- or late gestation, or to carry a high risk of fetal demise or life-threatening conditions at birth. FCIs also intend to improve neonatal and prenatal survival of the offspring, and limit lifelong morbidity and mortality [6,7,8,9]. Of interest, low maternal risks were reported with FCIs [10], but these interventions are currently limited to a small subset of CHDs (Table 1).

Thus, the limitations of surgical and medicinal treatments of cardiovascular diseases have stimulated the scientific and medical field to search for novel strategies. Consequently, human stem cells have emerged as a promising source of cells for cardiac regeneration in patients to complement current medical and surgical interventions, in addition to serving as cellular models to uncover the underlying mechanisms of CHDs [11,12]. To date, numerous preclinical and clinical studies have been performed to treat CHDs using embryonic stem cells (ESCs), induced pluripotent stem cells (iPSCs), and adult stem cells, such as cardiac progenitor cells and mesenchymal stem cells [13,14,15,16,17]. Although the outcomes of these trials remain largely unsatisfactory, the usage of autologous and allogenic stem cells to complement surgical interventions in infants with CHDs have recently shown positive results, particularly in patients with hypoplastic left heart syndrome (HLHS) [3,13,18,19,20]. To further reduce surgical interventions in patients, the tissue engineering research field is actively developing cell-seeded clinical patches that are able to either grow in synchronization with the cardiovascular structures or to be gradually replaced by the newly formed tissues of treated infants [3]. In addition to their differentiation abilities, stem cells secrete a wide range of bioactive molecules (secretomes) and release extracellular vesicles, including exosomes, which have promising potential in facilitating the repair and regeneration of damaged adult cardiac tissues and improving heart function [21,22,23,24,25,26,27,28,29,30,31,32,33,34,35,36]. The secretomes and exosomes released by various cells have significant roles in (cardiac) development and tissue repair, and hold promising potential toward reversing CHDs, or at least mitigating their severity [19,37,38,39,40,41].

The causes of CHDs are often multifactorial and may involve inherited genetic mutations, sporadic developmental errors, and/or environmental factors [42,43,44,45,46]. Mutations or other genomic abnormalities have been shown to occur in genes that play crucial roles in cardiac development (Table 2), including genes encoding for transcriptional and epigenetic/chromatin remodeling factors (such as NKX2.5, TBX5, GATA4, CITED2, TBX20, p300, and CBP), cell signaling and adhesion proteins (such as ACVR1, NOTCH1, and PDGFRA), and structural sarcomere proteins (such as MYH6, MYH7, and ACTC1) [47,48]. Chromosomal anomalies resulting in syndromic complications, such as Down syndrome (trisomy 21), Turner syndrome (monosomy X), and DiGeorge syndrome (22q11.2 deletion), are often associated with increased risk of CHDs [47,48].

**Table 1 jpm-13-01263-t001:** Clinical trials related to congenital heart diseases and in utero applications.

Clinical Trial Number	Type and Status	Condition	Intervention or Treatment	Population	Evaluation	Results
NCT03944837	Interventional Single-group assignment; Status: recruiting.	Severe Fetal CHD	Transient maternal oxygen gas administration during echocardio-graphic and MRI imaging.	Pregnant woman ≥18 years old, with fetal diagnosis of specific CHDs and intention of prenatal treatment.	Brain growth and maturation to birth, improvement of postnatal neurodevelopmental issues, identification of CHD types likely to benefit from chronic maternal hyperoxygenation.	Not available
NCT01736956	Interventional, prospective, non-randomi-zed clinical trial; Status: complete.	Aortic stenosis and evolving HLHS	Fetal aortic transuterine valvuloplasty, periventricular approach. Control group with standard prenatal and postnatal care.	Pregnant woman ≥ 16 years old, with a fetus with normal heart anatomy and severe aortic stenosis.	Safety and efficacy of in utero percutaneous balloon dilation of fetal aortic valve with severe stenosis determined by fetal mitral valve and left ventricle growth, survival, and neurodevelopmental status.	No results posted
NCT03147014	Interventional; Status: complete.	Fetal HLHS, atrial septal aneurysm, aortic coarctation.	Cardiovascular response to maternal hyperoxygenation in fetal CHDs. All singleton fetuses with CHDs at all gestational ages are eligible.	Pregnant women with a fetus harboring CHDs diagnosed at various gestational weeks.	Participants received 10–15 min hyperoxygenation, assessed for middle cerebral artery pulsation; myocardial diastolic function; flow patterns across tricuspid, mitral valves, and ductus venosus; changes in heart output and ratio of right–left ventricle flow; flow at the aortic isthmus if aortic coarctation.	No results posted
EudraCT 2016-003181-12 *	Prospective cohort study; Status: complete.	Pregnant women with fetus at risk of pulmonary hypoplasia due to VSD/AVSD.	Sonographic assessment of pulmonary vascular reactivity following maternal hyperoxygenation.	Pregnant woman ≥18 years old, with a fetus at risk for neonatal persistent pulmonary hypertension; non-pregnant control.	Fetal echocardiographic doppler within the first 48 h of life to assess pulmonary vasculature prior to and after maternal hyperoxygenation to predict development of neonatal pulmonary hypertension.	[49]

CHD—congenital heart disease; HLHS—hypoplastic left heart syndrome; VSD—ventricular septal defect; AVSD—atrioventricular septal defect. All data collected from clinicaltrials.gov, except (*) from clinicaltrialsregister.eu/, accessed on 13 August 2023.

Maternal/embryonic–fetal environmental factors, including maternal diabetes, medications taken during pregnancy, infections, alcohol or drug abuse, exposure to tobacco smoke, chemicals or toxins, may also increase the risk for CHDs [5,48]. Of interest, embryonic heart and placental development are interrelated and concomitant, sharing regulatory molecules and signaling pathways. Placental defects are also more common in pregnancies with CHDs, and the inadequate placental function can contribute to persistent cardiac defects postnatally [50,51,52].

Despite the complexity of embryonic heart formation and the high incidence of CHDs, cardiogenesis is a remarkably robust developmental process. Indeed, dysfunctions of more than one gene and/or environmental insults are often necessary to drastically impair heart development and lead to severe CHDs or death during gestation [53,54,55]. The resilience of this process certainly relies on the complex and evolutionary conserved gene regulatory network, which is orchestrated by key signaling pathways and transcription factors, that belong to families of proteins sometimes displaying overlapping or redundant functions [56]. Their robustness also depends on the existence of multiple sources of cardiac progenitors with functional redundancy, ensuring the production of heart cells even in the event of loss or defects in formation, expansion, and differentiation of certain progenitors [57,58,59]. For instance, despite ablation of over 50% of emerging First Heart Field (FHF) and Secondary Heart Field (SHF) cardiac progenitors at early stages of cardiogenesis, mouse embryos survived and developed into adult animals without any apparent cardiac abnormalities [58,60]. Therefore, the dysfunction or loss of cardiac progenitors and cardiomyocytes in mouse embryonic hearts can truly be compensated by alternative embryonic mechanisms, which include the expansion and migration of other unaffected cardiac cells [58,60,61,62,63].

Recent studies, using animal models and in vitro differentiation of human pluripotent stem cells, have also suggested that cardiac cell lineages are pre-determined during the early stages of development. Both positioning and functions of cardiac cells in the heart may be defined even before gastrulation, and orchestrated by multiple signaling pathways and environmental cues [57,64,65,66,67,68,69]. This idea is also supported by the ability of pluripotent stem cells to generate self-organizing cardiac organoids when provided with external cues from the extracellular matrix (ECM) and signaling molecules [70]. Moreover, in the absence of maternal tissues in vitro, human blastocysts attached to instructive supports and/or, stimulated by signaling molecules (such as WNT and NODAL/ACTIVIN), autonomously self-organize and originate structures resembling the proper embryo, with key landmarks of normal development, including bilaminar disc formation, primitive streak formation, lineage commitment, and extraembryonic annexes [71,72,73,74]. Therefore, future clinical strategies that would consistently provide correct developmental signals at early embryonic stages have the potential to compensate for defective mechanisms, thereby significantly reducing the incidence of CHDs.

Here, we review key molecular and cellular processes involved in early heart development. We also explore recent findings supporting the idea that the targeted delivery of biomolecules or exosomes during embryonic development, as early as the blastocyst stage, may compensate for faulty genes, signaling pathways, and cardiac progenitors, and potentially reduce the occurrence and/or severity of CHDs.

## 2. Navigating Early Mammalian Embryonic Heart and Placental Development

### 2.1. Placental Development

Placental and embryonic heart developments occur simultaneously, forming a placenta–heart axis, and sharing developmental pathways and common susceptibility to genetic defects [51]. Thus, the developing heart is highly vulnerable to early placental insufficiency. During early human embryogenesis, at embryonic days 4 to 5 (E4-5), the blastocyst attaches to the uterine wall. Trophoblast cells, encasing the inner cell mass that will originate the embryo proper, differentiate into inner cytotrophoblast cells, which are key for blastocyst implantation in the uterus, and stem cells that will also originate the outer syncytiotrophoblast [75,76]. The syncytiotrophoblast is responsible for the exchange of nutrients, gases, and waste products between the maternal and embryonic circulations, and also acts as a barrier against pathogens and harmful substances [77]. Interactions between the cytotrophoblast and syncytiotrophoblast are essential for proper placental development and function. Aberrant vascular placental development can lead to placental insufficiency and compromise fetal growth due to inefficient blood supply, and may result in preeclampsia, gestational diabetes, intrauterine growth restriction (IUGR), and placental abruption [78,79,80,81,82]. These complications are associated with an increased risk of adverse fetal outcomes, including CHDs [83,84,85,86]. Vascular endothelial growth factor (VEGF) and its receptors, hypoxia-inducible factors (HIFs), which regulate the cellular response to hypoxia, play a crucial role in placental and heart development, as well as in vascular remodeling, and their dysfunction increases the risk for CHDs [75,87,88,89]. Immune modulation is also essential for placental development and for fetal tolerance by the maternal immune system [90,91].

### 2.2. Heart Development

During embryogenesis, the heart is progressively built with multiple mature and functional cells, originating from many molecularly distinct progenitor cells that arise at different times and structures from the developing embryo [57,65,66,67,92,93]. The heart primarily develops from three main pools of embryonic progenitors, which are the FHF, SHF, and the proepicardium cells (Figure 1). The FHF and SHF derive from the earliest MESP1-marked cardiac mesodermal progenitors, which emerge shortly after gastrulation [57,64,65,66,67]. However, precardiac progenitor cells in the nascent mesoderm are, in fact, a heterogeneous population, composed of molecularly distinct progenitors that leave the primitive streak in a sequential manner, with contributions to specific and overlapping parts of the heart [93,94,95].

The first cells to leave the primitive streak, emerging at approximately E15-16 in humans, are FHF cells that give rise to the cardiac crescent structure, which migrates and fuses at the midline to form the primitive linear heart tube (at around E19 in humans) [42,57,64,65,66,93,96,97]. The linear heart tube, destined to generate the bulk of the atrial chambers and left ventricle, contains an outer myocardium and an inner endocardium, separated by a specific ECM known as the cardiac jelly. Rhythmic contractions are initiated by cardiomyocytes of the heart tube at around E21 in humans. At approximately E19-20, human SHF cells migrate into the heart tube and differentiate into cardiomyocytes, smooth muscle, and endothelial cells. The SHF contributes to the outflow tract (OFT), the right ventricular region, the atrioventricular canal structures, and the atria [42,57,64,65,66,93,96,97].

Around E28, the human heart tube undergoes a rightward looping to initiate the formation of the four-chambered heart. At this stage, the conductive system also begins to develop from cardiac progenitors, which differentiate into specialized myocytes with high conductance to form the sinoatrial node, atrioventricular node, bundle of His, and Purkinje fibers, rather than working cardiomyocytes [98,99]. As the heart develops, conduction cells establish connections to form a functional electrical system that coordinates the contraction and relaxation of the cardiac muscle. Further maturation and refinements of the conductive system occur throughout fetal development, with further structural and functional changes happening after birth and during postnatal growth [98,99]. In humans, ventricular septation starts around E50 and originates from the myocardium, which generates the ventricular septum and establishes the right and left atrio-ventricular canals. Atrial septation starts at E60 and derives from the septum primum and septum secundum, concomitantly with OFT septation [42,67,100,101]. 

OFT septation, which results in the formation of the aorta and the pulmonary artery, and their respective connections to the left and right ventricles, is accomplished by SHF cells and cardiac neural crest cells (CNCCs), a subpopulation of neural crest cells that delaminate from the neural tube [101,102]. CNCCs also give rise to the tip of the interventricular septum and OFT cushions, which will differentiate into the aortic and pulmonary valves and the parasympathetic coronary innervation [102]. The proepicardial organ is an extracardiac structure, which develops distinctly from the heart tube (at E9.5 in mouse) and contributes to the epicardial cells located around all heart chambers [95,103,104]. Other early multipotent progenitor cells, marked by KDR/FLK1/VEGFR2 expression, which are a source for endocardium, myocardium, and hematopoietic progenitors, were also detected in the primitive streak [57,105]. The major functional structures of the human heart are completed by E60. After this stage, the heart undergoes progressive growth, as well as structural, metabolic, and functional maturation processes, which are vital for its function during a lifetime [70].

## 3. Tracing the Roles of Signaling Pathways and Exosomes in Heart Development

### 3.1. Secretomes and Heart Development

Secretomes refer to a variety of secreted proteins, bioactive molecules, and factors released by cells that act locally to modulate cellular processes on neighboring cells, including processes necessary for development and tissue repair [30,106]. Among those factors, modulators of the WNT signaling pathway play a critical role in cardiac specification and differentiation, through stabilization and nuclear translocation of β-catenin, which, in turn, regulates the expression of cardiogenic genes [107,108,109,110,111,112,113]. NOTCH receptors and ligands are also important cardiogenic mediators through cell–cell communication, controlling the differentiation of cardiac progenitor cells into various cardiac lineages during heart development [114,115,116,117,118,119,120,121,122,123,124,125,126]. Bone morphogenetic proteins (BMPs), which are key regulators of embryonic development, promote the differentiation of cardiac mesoderm and regulate the formation of heart structures, through the expression of cardiac-specific transcription factors critical for cardiomyocyte differentiation [127,128,129,130,131,132,133,134,135,136,137]. The fibroblast growth factor (FGF) pathway is important for the formation of the OFT, ventricular maturation, and valve development, by modulating cell proliferation, survival, and differentiation [138,139,140]. Other secreted factors, such as insulin-like growth factor 1 (IGF-1) [141,142,143,144,145], VEGF [146,147,148,149,150], and transforming growth factor beta (TGF-β) family members, including ACTIVIN and NODAL [115,151,152,153,154], have been shown to enhance cardiomyocyte differentiation and maturation. Both ACTIVIN and NODAL crosstalk with other signaling molecules and modulate the activity of transcription factors to coordinate the specification, proliferation, and differentiation of cardiac progenitors during heart development (Figure 1B). Gastrulation is a critical embryonic step resulting in the formation of the three germ layers (endoderm, mesoderm, and ectoderm) and the specification of cell fates. During this process, mesodermal induction is stimulated by the convergent action of BMPs, NODAL/ACTIVIN, and canonical WNT signaling [93,155,156]. The emergence of mesodermal cells is marked by the expression of the transcription factor BRACHYURY and requires the activation of β-catenin by the canonical WNT pathway. Next, the inhibition of the WNT canonical pathway, achieved by proteins such as DKK1 and noncanonical WNT-related proteins, is critical for cardiac mesoderm specification [109,113,156,157,158]. Angiogenesis and vasculogenesis are also critical for cardiovascular development, as they ensure the formation of functional vascular networks through which to supply oxygen and nutrients to the developing heart. These processes are modulated by secreted proteins, such as VEGF, basic FGF, and angiopoietin-1, which are also capable of enhancing neovascularization in adult ischemic hearts [159,160,161,162].

### 3.2. Exosomes and Heart Development

Exosomes are formed through the inward budding of endosomal membranes and are subsequently released into the extracellular space to be captured by target cells [163]. The proteomic analysis of exosomes has revealed the presence of proteins involved in cell signaling, membrane transport, and ECM remodeling [164]. Additionally, exosomes may contain various types of messenger RNA (mRNA), microRNA (miRNA), long non-coding RNA (lncRNA), and mitochondrial miRNA [165,166,167], which can be transferred to recipient target cells and alter their gene expressions and cellular functions. During heart development, exosomes from multiple cell sources, including stem cells, have emerged as key mediators of intercellular communication in various biological contexts, including cardiac regeneration or repair (Table 3) [37,168,169,170]. For example, cardiac exosomes and secreted factors derived from cardiomyocytes, endothelial cells, or cardiac progenitor cells are important to the regulation of cardiac cell fate determination, proliferation, migration and differentiation, and heart morphogenesis [24,30,166,171,172,173]. Moreover, exosomes derived from cardiac and endothelial cells modulate epithelial-to-mesenchymal transition, favoring angiogenesis and the formation of coronary vessels, which are essential for heart development [174,175,176,177]. In addition, exosomes released through circulation during pregnancy via the umbilical cord, placenta, amniotic fluid, and amniotic membranes contribute to essential physiological functions in fetal–maternal communications, and to physiological processes such as angiogenesis, endothelial cell migration, and embryo implantation and placentation [178,179,180].

Some exosomes, such as those derived either from diabetic or obese pregnant mice, cross the maternal–fetal barrier when injected through circulation and contribute to anomalies in cardiac development, such as ventricular septal defects, cardiac hypertrophy, pericardial effusions, and compromised systolic and diastolic functions [178,181,182,183]. Exosomes from the visceral adipose tissue of obese pregnant mice displayed an altered composition in miRNA, such as a decrease in miR-19b, which is involved in regulating inflammation and cardiac development, resulting in altered placental and cardiac functions [183]. Circulating exosomes, which are characteristic of maternal metabolic adaptations, have also been proposed to serve as pregnancy complication markers for gestational diabetes, hypertension, and pre-eclampsia, and to facilitate the determination of embryos with neural tube defects, CHDs, and other conditions, such as Down syndrome [178,184,185,186]. Overall, paracrine signaling mechanisms, mediated by exosomes and secreted factors, ensure the global regulation of cardiac morphogenesis [187], and their dysregulation is likely to result in CHDs [157,188,189,190,191].

## 4. Exploring the Impact of Diet and Medicinal Supplements on Heart Development

### 4.1. Diet Supplements and Heart Development

Both animal model studies and human trials have shown promising outcomes in using drugs and dietary supplementation as preventive measures for CHDs, setting the possibility to interfere with adverse cardiac defects through the intake or administration of biomolecules. Folic acid supplementation before and during pregnancy is a notable example, as it reduces the risk of neural tube defects and certain CHDs, particularly septal defects [192,193]. Supplementation of zinc (Zn) to diabetic mothers has also effectively reduced glucose-dependent oxidative stress, associated with cardiomyopathy in animals and humans, by preventing apoptosis in cardiomyocytes, and its supplementation at early stages of pregnancy was proposed to prevent CHDs induced by gestational diabetes [194]. Retinoic Acid (RA), a vitamin A derivative, is indispensable for many embryonic developmental processes, including heart formation (Figure 1), but the rigorous control of its production by cardiac embryonic tissues is necessary to prevent disastrous congenital malformations, such as anomalies in heart looping, the aortic arch, transposition of the great arteries, coronary defects, double-outlet right ventricle, myocardial hypoplasia, tetralogy of Fallot, and OFT and septal defects [195]. The injection of wild-type ESCs into blastocysts deficient in RA signaling successfully rescued various morphogenetic anomalies, such as left–right heart tube looping defects, through increased production of RA, which diffused to activate RA signaling in the defective cells [196]. This study demonstrated that endogenous embryonic production of RA can compensate for developmental heart defects. However, maternal supplementation was unable to provide the necessary levels of RA needed to rescue the mutants [196]. These examples indicate the possibility to limit CHDs by supplying biomolecules to mothers or embryos in pre- or early pregnancies. However, excessive fetal exposure to 13-cis-RA supplements (isotretinoin, Accutane) can induce CHDs. Alterations in fetal retinoid metabolism due to cross reactions with drugs (valproate), toxins (nitrofen, tobacco, alcohol), infections (rubella), or coexisting medical conditions (gestational diabetes) also favored CHDs [195]. Therefore, to prevent CHDs, it is crucial to have a thorough understanding of the roles and optimal levels of vitamin A, as well as of the factors that influence retinoid metabolism [197]. In general, a careful determination of the intake quantity of dietary supplements or drugs will be crucial to developing safe strategies for promoting normal heart development, achieving cardio-protection, and avoiding adverse effects, including CHDs themselves.

### 4.2. Cardiac Complications Due to Drugs Used to Treat Non-Cardiac Defects

Particular attention needs also to be given to supplements used for the treatment of non-cardiac problems, such as prenatal corticosteroid therapies used to stimulate fetal lung maturation in preterm births, which may affect fetal heart development and function in adulthood. Indeed, some studies in animal models have suggested that fetal exposure to glucocorticoids may cause changes in cardiomyocyte size and collagen deposition, and impair cardiac contractility, ultimately leading to an increase in the left ventricular mass and an alteration in cardiac function [198]. Maternal steroid therapy used to treat or prevent fetal heart block has also been shown to have undesirable side effects on the fetuses, such as alterations in brain development, oligohydramnios, growth restrictions, and constriction of the arterial duct, as well as diabetes mellitus, adrenal insufficiency, and psychosis for the mother [199,200]. Overall, several supplements administered before and during early pregnancy have shown potential for preventing CHDs and other congenital conditions. However, more research is required to determine the appropriate dosage of supplements in order to maximize the benefits for heart development while minimizing any adverse effect.

## 5. Unveiling the Potential of Secretomes and Exosomes for CHD Prevention

### 5.1. Exogenous Instructive Molecules to Mitigate CHDs

Multiple signaling pathways, which may display overlapping or redundant functions, are involved in generating cardiac progenitors and other cell lineages [57,64,65,66,67,68,69]. All issues considered, it is tempting to propose that the supplementation of exogenous cardiogenic instructive molecules at key moments during early embryonic stages may compensate for defective pathways and/or lineage decisions of cardiovascular progenitors to mitigate CHDs. To date, only a few experimental reports on animal models and stem cells support this idea. For instance, prolonged hypoxia causes DNA damage, premature senescence, impaired angiogenesis, and fibrosis associated with the upregulation of TGF-β1 expression in human fetuses with HLHS [201]. In vitro, hypoxia exposure of a human HLHS-derived iPSC disease cell model promotes the differentiation into cardiac fibroblasts instead of cardiac progenitors, cardiomyocytes, and endothelial cells [201]. Inhibition of TGF-β1 activity by its antagonist compound SB431542, in HLHS-derived iPSC exposed to hypoxia, prevented senescence and promoted genomic stability [201], suggesting that early interventions to inhibit TGF-β1 could improve ventricular growth and overcome pathways dysregulated in HLHS. IUGR is another gestational defect that hinders fetal growth in the uterus. IUGR is associated with increased placental vascular resistance, which forces the workload onto the fetal heart and increases the risk of cardiovascular disease [202]. Maternal administration of IGF-1 and IGF-2, which are known to stimulate placental and fetal growth in animal models, showed promising potential to treat IUGR in cases where downregulation of IGF-1 receptors in the placenta is not observed [202,203].

### 5.2. Exosomes, a Non-Cellular Approach for Correcting Embryonic Cardiac Defects

Emerging evidence suggests that the beneficial effects of stem cells in medical applications may primarily result from paracrine signaling molecules and the release of extracellular vesicles, rather than from the direct integration of stem cells or their derived cells into the target tissue [204]. Exosomes, which can be isolated and characterized by well-established approaches [205,206], present advantages such as scalable production, easy storage, consistent morphology and function, compliance with regulatory standards, and possible reduction in the variability of outcomes associated with cell therapies [207]. The therapeutic potential of exosomes and secretomes derived from hormonally primed human endometrial epithelial cells was tested in mouse models, and proven to enhance embryonic growth, development, and implantation [181]. This study also suggested that dysfunctions of the endometrial secretomes, which hinder implantation in cases of infertility, can be amended through the supply of exogenous exosomes. Also, maternal diabetic pregnancies in mouse models showed a higher occurrence of neural tube defects associated with exosomes produced from vascular progenitor cells expressing FLK1 derived from mesoderm cells lacking Survivin [208]. Interestingly, delivery into the amniotic cavity of Survivin-enriched vascular progenitor exosomes prevented neural tube defects in diabetic pregnancies [208], demonstrating the capacity of modified exosomes delivered in utero to limit neuronal pathology. Thus, exosomes are a promising non-cellular alternative to cells for clinical applications to correct embryonic defects. Exosomes can also be engineered to carry specific cargo, such as miRNAs or even gene-editing tools more suitable for the correction of genetic abnormalities associated with CHDs [176,209]. Notably, exosomes derived from the cardiac progenitors of children undergoing reconstructive heart surgeries showed promising effects in promoting angiogenesis, reducing fibrosis, resolving hypertrophy, and improving cardiac function in a rat model of heart arrhythmias caused by ischemia-reperfusion injury [39], showing the cardioprotective effect of exosomes from infant cardiac cells. Interestingly, exosomes from neonatal progenitors improved cardiac function regardless of oxygen levels, while exosomes from older children were only reparative in hypoxic conditions. Therefore, further research is needed to determine the most suitable cell sources and culture conditions for the production and modification of exosomes that would yield the best personalized clinical outcomes in the treatment of CHDs [40,208]. 

## 6. Harnessing Secreted Factors in Embryogenesis for Protection against CHDs

Like in many diseases, the early correction of a CHD is likely to result in a better clinical outcome, through early normalization of cardiogenesis. Reports on stem cell differentiation and heart development have suggested that cell positioning and functions within the heart may be determined prior to gastrulation, through precise signaling pathways and environmental cues [57,64,65,66,67,68,69]. Therefore, early interventions aiming to target the blastocyst at the pre-gastrulation/pre-implantation stage, very early embryonic states, may reduce the occurrence and severity of CHDs. Interestingly, pioneering studies on *Inhibitors of Differentiation* (*ID*) genes have revealed that double knockout of any pair of these genes in mouse models results in prominent cardiac defects and midgestational lethality of the embryos [210,211]. However, the injection of wild-type ESCs, either in ID-null blastocysts to form a chimera, or intraperitoneally in females prior to conception, rescued cardiac malformations and viability of ID-null embryos through upregulation of IGF-1 and WNT5A, without incorporation of wild-type ESCs into ID-null embryos [210,211]. Similarly, a chimeric embryo formation, using wild-type ESCs, improved the morphogenetic defects of RA signaling-deficient embryos by increasing RA expression [196]. Interestingly, transient exogenous WNT5A and WNT11, supplied at the 1-cell stage, efficiently rescued the viability and cardiac defects of CITED2-depleted zebrafish embryos and cardiac differentiation of mouse ESCs [212]. Dysfunctions of CITED2, a transcriptional modulator, have been widely associated with zebrafish, mouse, and human CHDs, as well as with embryonic lethality [55,212,213,214,215,216,217,218,219,220]. At the cellular level, CITED2 regulates the expressions of numerous genes involved in early cardiogenesis, such as BRACHYURY, MESP1, ISL1, GATA4, TBX5, MEF2C, NODAL, LEFTY1/2, PITX2C, VEGFA, WNT5A, and WNT11, among others [212,213,214]. Thus, the supplementation of WNT5A and WNT11 at the blastocyst stage in utero holds potential for rescuing CHDs associated with CITED2 dysfunction in mammals, since the function of CITED2 is conserved across vertebrates [212,214,216,221]. This strategy may also prevent CHDs triggered by the dysfunction of many genes other than *CITED2*, such as CITED2-target cardiogenic genes, including WNT5A and WNT11.

Moreover, WNT5A and WNT11 trigger many congenital cardiac anomalies, and contribute to DiGeorge syndrome, when defective [222,223,224,225,226,227,228]. Indeed, the WNT5A and WNT11 proteins are central for proper gestation and successful pregnancy, and their expressions are naturally highly increased and localized in the uterine luminal epithelium prior to and during blastocyst attachment to the uterus [229,230]. These proteins also promote embryonic uterine implantation and survival [229], as well as placental growth [230]. WNT5A and WNT11 are also important in late gastrulation, for the regulation of the anterior–posterior axis elongation, notochord extension, and proper patterning of the neural tube and somites [231]. In early mouse gastrulation, WNT11 is initially expressed in endoderm progenitors, and later, during mid-gastrulation, it plays a role in the formation of the embryonic and extraembryonic endothelia, as well as the formation of the endocardium in all chambers of the developing heart [232]. During late gastrulation, WNT11 showed successive waves of expression in different regions of the myocardium, important to originate left ventricle precursors (FHF progenitors) from E7.0–8.0, right ventricle progenitors (SHF progenitors) from E8.0–9.0, and the superior wall of the OFT from E8.5–10.5 (also SHF progenitors) [232]. Other studies have implicated WNT5A and WNT11 in the development of cardiac progenitors in vitro and in vivo [233,234,235,236], proper fetal hematopoiesis [237,238], kidney development [239], and guidance of sympathetic neurons to their innervation targets in vivo [240], among other processes. Altogether, the broad range of effects exhibited by WNT5A and WNT11 during early development emphasizes their high potential in limiting CHDs and other birth abnormalities when supplied exogenously at early embryonic stages. Another recent study highlighted the role of a subset of human amniotic epithelial cells (AECs) in mesoderm formation at E8.0–9.0 during early post-implantation stages [241]. Interestingly, impaired mesoderm formation and lethality due to loss of ISL1 expression in non-human primate embryos, or in human AEC differentiation in vitro due to the decrease in BMP4 expression, was partially restored through supplementation of BMP4 [241]. Overall, these findings demonstrate the potential for biomolecules, such as WNT5A/WNT11, BMP4, and inhibitors of the TGF-β pathway (SB431542, for instance), to reduce CHDs when administered at very early stages of development.

## 7. Challenges and Ethical Considerations

The experiments on mouse ID-null and RA-defective embryos rescued by the introduction of ESCs, either in defective blastocysts or in the mother prior to gestation [196,210,211], are groundbreaking studies, establishing a proof-of-principle that early interventions have the potential to restore normal development to genetically defective blastocysts. However, the translation of such strategies to humans, based on pluripotent stem cell delivery into blastocysts, remains both ethically and technically undesirable. Indeed, the alteration of such early human embryos, even for health purposes, is per se questionable. Moreover, the transplantation of human pluripotent stem cells (ESCs or iPSCs) in a clinical setting carries high risks of tumor formation and differentiation into unwanted cells/tissues [242]. Nevertheless, pre-implanted blastocysts are promising targets for medical interventions, since their size and localization, near or in the uterine lumen, may facilitate the delivery and effective diffusion of corrective biomolecules. Moreover, blastocysts exhibit minimal tissue commitment bias, making them possibly more responsive to exogenous cues aiming to correct congenital defects. In both studies presented above, the corrective effects of wild-type ESCs were contributed by the neomorphic effects of factors secreted by these cells, such as WNT5A, IGF-1, and RA [196,210,211]. These findings strongly support the idea that exogenous administration of WNT5A and WNT11, as well as IGF-1 and possibly other factors to be identified, to early embryos has the capacity to substitute for pluripotent stem cell transplantation.

The intricate expression patterns and multiple functions of the WNT5A, WNT11, IGF-1, BMP4, and TGF-β pathways during placental and embryonic development portend that any disturbance in their expression and signaling network may have drastic impacts on normal embryonic development. However, the supplementation of these factors may only need to be temporary and administered at very specific times, as remarkably suggested by the transient supply of exogenous WNT5A/WNT11 recombinant proteins at the onset of mouse ESC differentiation for 2 days, or at the 1-cell stage in zebrafish embryos to rescue the cardiogenic defects and lethality caused by CITED2 depletion [212]. Moreover, compelling evidence suggests that the exogenous administration of WNT5A and WNT11 proteins to early embryos may not only be well-tolerated, but may even provide general beneficial effects on development, rather than potential adverse effects. Indeed, WNT5A is naturally and abundantly expressed in the endometrium of mammals during the pre-implantation period, ensuring proper placental and embryonic development from the morula to the blastocyst stage, safeguarding the success of pregnancies [229,230,243,244,245,246,247]. Therefore, the supply of exogenous WNT5A to the uterine lumen before implantation coincides with its naturally high expression during this period, suggesting that the organism may be prepared to handle abundant levels of WNT5A and may mitigate any potential adverse effects caused by its exogenous administration. In addition, WNT5A and WNT11 form protein complexes, which were proposed to be more effective in activating their cellular functions than the individual proteins were [248]. Thus, the combined administration of WNT5A/WNT11 may be more inclined to fulfil the rescuing effects and buffer the adverse effects. Accordingly, the combined supplementation of WNT5A/WNT11 was tendentially more efficient for rescuing the viability and cardiac anomalies of CITED2-deficient zebrafish embryos, and for restoring the expression levels of early cardiogenic factors (such as BRACHYURY, MESP1, and ISL1) to control levels in CITED2-depleted mouse ESCs [212]. Also of interest, the combined administration of WNT5A/WNT11 to wild-type zebrafish embryos additionally reduced natural occurrences of death, cardiac anomalies, and variability in heart rate, rather than showing adverse effects [212]. Together, these observations suggest that the temporary supply of WNT5A/WNT11 in the uterine lumen during peri-implantation, to target the blastocyst, would meet tolerant conditions, bearing increased WNT5A/WNT11 levels able to prevent CHDs with no or minimal side effects. Further investigation is needed to determine whether other cardiogenic molecules, such as BMP4 or antagonists of TGF-β, IGF-1, VEGF, and HGF, for example [106,160,249,250,251], as well as relevant exosomes, would also be able to limit CHDs, if delivered at the pre-implantation period.

Given the multifaceted nature of CHDs, and the intricate cellular interactions involved in cardiogenesis, there is potential for synergistic and amplified therapeutic effects by combining approaches, such as exosomes/secretomes and drug delivery, with gene therapy and/or tissue engineering. Furthermore, personalized medicinal strategies that consider individual genetic and molecular profiles would help to tailor treatments and improve care for patients with specific genetic factors associated with CHDs [28,252,253]. Future challenges also include standardization and optimization of secreted protein/exosome production, and full comprehension of their complex interplay with recipient cells to translate preclinical findings into safe clinical applications [254,255,256].

Finally, the reconsideration of ethical guidelines by the pillars of ethical governance, to shape responsibility and trust, is necessary to avoid the offer of defiant, unsafe, and unproven treatments by unauthorized clinics, similar to the existing issue in the field of stem cell-based therapies [257,258,259]. Indeed, the excitement and expectations surrounding such novel medical approaches may lead individuals at high risk of having children with CHDs to seek unapproved therapies, invoking the “right to try” concept. Therefore, balancing individual autonomy with regulatory oversight is essential to protecting patients’ well-being. Additionally, equitable access to these potential treatments should be ensured, regardless of socio-economic status, geographic location, or other discriminative barriers. Lastly, a responsible approach to research, including clinical trials, is necessary for the future use of these approaches when no alternative options are available, aligning with the ethical principle of proportionality.

## 8. Conclusions and Perspectives

Mammalian heart development is a complex process, involving distinct pools of cardiac progenitor cells located in specific regions of the embryo. The precise orchestration of the progenitor cells’ emergence in embryonic time and space, as well as their proliferation, migration, and differentiation process, which is stimulated by environmental signals and pathways, is crucial for proper embryonic heart assembly. Despite high complexity, the cardiac developmental process remains robust, probably due to overlap and redundancy of the main signaling pathways, and of the multiple cardiac progenitors driving cardiogenesis. However, the dysfunction of cardiac progenitor cells, and/or the disruption of molecular mechanisms underlying cardiogenesis, results in cardiac malformations in 1 out of 100 human births, and often in embryonic death during gestation. Despite prenatal screening programs and better genetic counselling, which play a crucial role in preventing these malformations, CHDs remain the most prevalent congenital anomalies worldwide.

The current therapeutic approaches for severe CHDs, which are primarily surgical, remain unsatisfying despite tremendous medical and technological progress. Indeed, interventions are mostly palliative and often result in a lifelong health burden for patients. Moreover, patients are typically treated for the first time during the peri- or prenatal period, when the heart has fully developed from deficient early embryonic processes. Thus, earlier interventions in embryonic development could mitigate the impact of faulty heart development and reduce the occurrence of congenital abnormalities. Only recently, FCIs have been implemented for a reduced subset of CHD cases linked to hypoplastic left heart, to correct cardiac malformations in utero during pregnancy. FCIs have yielded promising results, and, compared to conventional surgical interventions after birth, FCIs offer the advantage of intervening at a critical developmental stage, when the heart is still growing and developing. These early corrections aim to address the underlying issues and abnormalities in cardiac structure and function before they become more severe, or irreversible.

Decades of cardiovascular research, encompassing studies on animal and cell models, as well as human genetic investigations, have uncovered the fundamental mechanisms that drive heart development and CHDs. Importantly, these findings have revealed the existence of multiple cardiac progenitors, as well as key transcription factors and signaling pathways involved in proper heart development. The cardiogenic process is characterized by overlapping and redundant functions existing among cardiac progenitors, signaling pathways, and factors. Redundant features in cardiac development enable the heart to sometimes compensate for defects in progenitors or genetic abnormalities through alternative mechanisms. Thus, despite the imperfect nature of the cardiogenic system, most viable individuals have a functional heart with minimal defects that can sustain them from gestational life through adulthood. Most importantly, these findings highlight the potential to artificially activate embryonic genetic and environmental cues to overcome defective cardiogenic events and compensate for adverse heart developmental conditions in utero.

Recent reports have emphasized the potential of specific biomolecules, including WNT5A, WNT11, IGF-1, BMP4, and exosomes, to interfere with abnormal cardiac development and correct cardiogenesis, when administered exogenously at early stages of development. Among those, WNT5A and WNT11 are particularly interesting, since they have shown corrective properties in mouse and fish models. WNT5A and WNT11 are necessary for correct development of the heart and placenta, among other organs and tissues, and they are abundantly expressed in the uterine epithelium at peri-implantation, a key period when blastocysts reach the uterine lumen. In addition, the blastocyst may be the target of choice for WNT5A and WNT11 corrective effects, since these early embryonic structures may be responsive to the exogenous instructions necessary to originate functional pre-gastrulation cells/progenitors, defined to derive specific parts of heart structures. Thus, it is of interest to fully explore the potential of WNT5A and WNT11 for the development and implementation of future preventive therapies for CHDs.

Nevertheless, further understanding of developmental mechanisms, as well as dosages, functions, and cross-reactions of therapeutic molecules, is key to devise the best and safest approaches to use all biomolecules and proteins (including WNT5A, WNT11, IGF-1, and BMP4), as well as exosomes, for CHD prevention. Additional research on genetic and environmental cues, as well as the mechanisms overcoming defective cardiogenic events, may help to identify other molecules and exosomes and develop safe strategies through which to compensate for adverse heart developmental conditions. Implementing such non-surgical interventions during early gestation offers the potential for reducing the occurrence and severity of CHDs, as well as an early preventive measure in utero. Although the underlying genetic mechanism potentially leading to CHDs would not be addressed with those approaches, it would be of interest to explore such combinatorial strategies, since they may provide prophylactic options to prevent or limit the severity of CHDs.

## Figures and Tables

**Figure 1 jpm-13-01263-f001:**
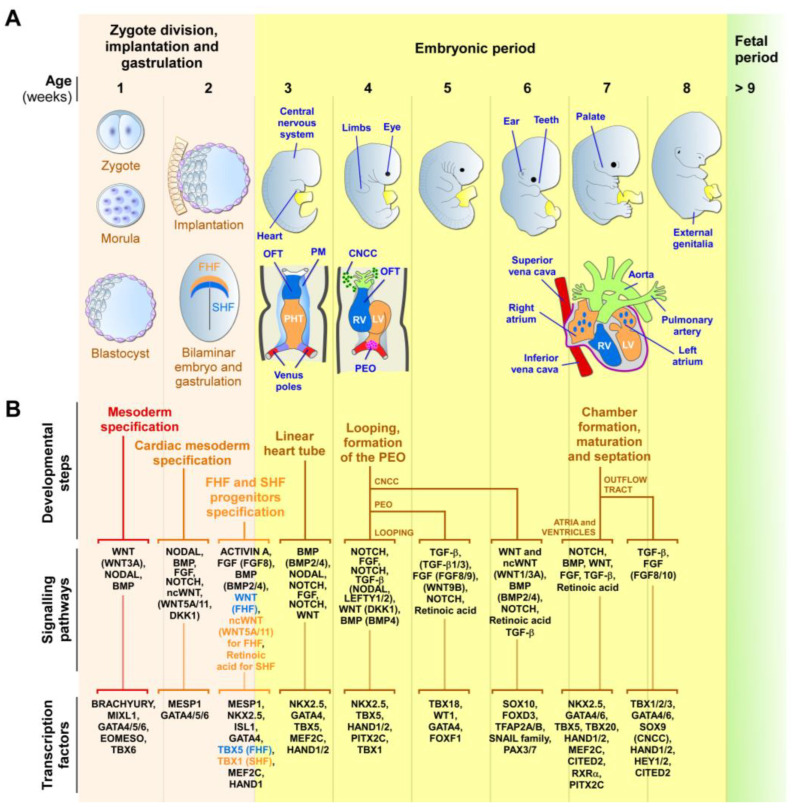
Early human cardiac development. (**A**) Schematic representation of the zygotic and embryonic stages of human development, highlighting key developmental programs and milestones. Gestational times are indicated in weeks. The initial steps of heart, central nervous system, limb, eye, ear, tooth, and genital development are shown. The primary (FHF—blue) and secondary (SHF—orange) heart field cells, the cardiac neural crest cells (CNCCs—green), and the proepicardial organ (PEO—purple), and their respective derivatives, are illustrated. During week 2, the cardiac mesoderm gives rise to progenitors of the FHF and SHF, which form the cardiac crescent. The FHF originates the primary heart tube (PHT), which subsequently contributes to the left ventricle (LV) and parts of the right and left atria. SHF cells migrate through the pharyngeal mesoderm (PM) and contribute to the elongation of the PHT by ingression at both atrial and venous poles. SHF cells contribute to the development of the right ventricle (RV), outflow tract (OFT), atria, and inflow myocardium. Cells originating from the venous poles (red) give rise to the superior and inferior vena cava. The PEO cells contribute to the epicardium and coronary vessels. CNCCs migrate from the dorsal neural tube into the cardiac OFT, where they contribute to the formation of the septum, separating the truncus arteriosus into the aorta and pulmonary artery, as well as contributing to heart valve formation and parasympathetic innervation. (**B**) Overview of the secretomes, main signaling pathways, and transcription factors that regulate each listed stage of heart development. The noncanonical WNT pathway is indicated as ncWNT.

**Table 2 jpm-13-01263-t002:** Genes frequently mutated and associated with congenital heart diseases.

Gene	Main Function in Cardiac Development	Cardiac Phenotype
Genes Encoding for Transcriptional and Epigenetic/Chromatin Remodeling Factors		
*ANKRD1*	Regulation of cardiac gene expression, muscle growth, and heart tissue maturation.	TAPVR
*CBP **	Transcriptional co-activator regulating the expression of genes critical for heart development, histone acetyltransferase and chromatin remodeling factor, essential for cardiac differentiation processes.	ASD, CoA, HSLS, MVD, PFO, PLSVC, vascular ring, VSD
*CITED2*	Regulation of cardiac gene expression, cardiac chamber formation, establishment of left–right asymmetry, heart and outflow tract septation.	AS, ASD, AVSD, PDA, PS, RAA, TGA, TOF, VSD
*FOG2/ZFPM2*	Stimulation of cardiac chamber formation and proper cardiomyocyte differentiation.	DORV, TOF
*FOXH1*	Forkhead activin signal transducer regulating NODAL signaling to specify the left–right axis and promote proper cardiac morphogenesis and septation.	TGA, TOF
*GATA4*	Regulation of cardiac gene expression, cardiac cell differentiation, and chamber formation.	ASD, AVSD, PAPVR, PS, TOF, VSD
*GATA5*	Cardiac cell fate determination and differentiation, and regulation of chamber-specific gene expression.	Bicuspid aortic valve, VSD
*GATA6*	Regulation of cardiac cell differentiation, heart morphogenesis, and septation.	ASD, AVSD, OFT defects, PDA, PS, TOF, VSD
*NKX2.5*	Regulation of cardiomyocyte differentiation, cardiac chamber formation, and establishment of the electrical conduction system.	ASD, CoA, DORV, HSLH, IAA, OFT defects, TGA, TOF, VSD
*P300 **	Transcriptional co-activator regulating the expression of genes critical for heart development, histone acetyltransferase and chromatin remodeling factor, essential for cardiac differentiation processes.	AVD, congenital aortic aneurysm, MVD, PDA, PS, TOF, VDS
*TBX1*	Regulation of cardiac progenitor cells, conduction system formation, and outflow tract morphogenesis.	TOF, 22q11 deletion syndrome
*TBX5*	Control of cardiac cell fate, chamber formation.	ASD, AVSD, VSD, Holt–Oram syndrome
*TBX20*	Control of cardiac cell differentiation, chamber formation, and cardiac gene expression patterns.	ASD, MVD, VSD
*TFAP2B*	Cardiac neural crest migration, outflow tract septation, and aortic arch patterning.	PDA, Char syndrome
Cell signaling and adhesion proteins		
*ACVR1/ALK2*	Receptor for BMPs (TGF-β signaling pathway family) controlling cardiomyocyte differentiation, valve formation, and heart morphogenesis.	AVSD
*ACVR2B*	Receptor for various ligands of the TGF-beta family, such as activins, myostatin, and growth and differentiation factors (GDFs), modulating signaling pathways involved in cardiomyocyte differentiation, cardiac morphogenesis, and chamber formation.	Dextrocardia, DORV, PS, TGA, TOF
*CFC1*	Co-receptor for NODAL contributing to left–right patterning and cardiomyocyte differentiation.	ASD, AVSD, DORV, IAA, TGA, TOF, VSD
*GJA1*	Promotion of the electrical coupling between cardiomyocytes, and contribution to proper cardiac conduction and rhythm establishment.	ASD, HLHS, TAPVR
*JAG1*	NOTCH ligand important to regulate cardiomyocyte differentiation, cardiac chamber formation, and valve morphogenesis.	PAS, TOF, Alagille syndrome
*LEFTY2*	NODAL inhibitor playing a role in left–right patterning, cardiac morphogenesis, septation, and chamber formation.	AVSD, CoA, IAA, IVC defects, Left–Right axis defects, TGA
*NODAL*	Control of left–right axis determination and promotion of cardiac looping, chamber formation, and valvulogenesis.	AVSD, Dextrocardia, DORV, IVC defects, PA, TAPVR, TGA, TOF
*NOTCH1*	Regulation of cardiomyocyte differentiation, cardiac valve formation, cardiac cell fate, and cardiac cell maturation.	AS, BAV, CoA, HLHS
*PDGFRA*	Participation in cardiac neural crest migration, cardiomyocyte proliferation, and proper outflow tract formation.	TAPVR
*VEGF*	Promoting angiogenesis and vascularization, ensuring proper blood supply to developing cardiac tissues.	CoA, OFT defects
Structural sarcomere proteins		
*ACTC1*	Encoding the major protein component of cardiac muscle, contributing to the formation and contraction of cardiac tissue.	ASD
*MYH6*	Encoding a major contractile protein in cardiac muscle fibers, contributing to proper heart contraction and function.	AS, ASD, HLHS, PFO, TGA
*MYH7*	Encoding a major contractile protein in cardiac muscle, playing a key role in cardiac contraction and function.	ASD, Ebstein Anomaly, NVM

AS—aortic stenosis; ASD—atrial septal defect; AVSD—atrioventricular septal defect; BAV—bicuspid aortic valve; CoA—coarctation of the aorta; DORV—double-outlet right ventricle; HLHS—hypoplastic left heart syndrome; IAA—interrupted aortic arch; IVC—inferior vena cava; MVD—mitral valve defect; OFT—outflow tract; NVM—noncompaction of the ventricular myocardium; PAPVR—partial anomalous pulmonary venous return; PDA—patent ductus arteriosus; PFO—patent foramen ovale; PLSVC—persistent left superior vena cava; PS—pulmonary (valve) stenosis; RAA—right aortic arch; TAPVR—total anomalous pulmonary venous return; TGA—transposition of the great arteries; TOF—tetralogy of Fallot; VSD—ventricular septal defect. (*) In the context of the Rubinstein–Taybi Syndrome.

**Table 3 jpm-13-01263-t003:** Examples of miRNA enriched in exosomes from various human cell sources, and their effect on the cardiovascular system.

Cell Source	Active miRNA	Effect on Cardiovascular System	Disease/Experimental Model
Human bone marrow MSC	miR-22	Cardioprotective during ischemia.	LAD ligation, in vitro
	miR-19a	Cardiomyocyte survival and preservation of mitochondrial membrane potential.	LAD ligation, in vitro
	miR-144	Reduced apoptosis in embryonic rat cardiomyocytes.	Cell line, in vitro
Human ESC-derived MSC	miR-21	Reduced infarct size in mouse model.	Myocardial ischemia-reperfusion injury
Human heart stromal cells derived from healthy individuals	miR-21	Prevention of the left ventricular ejection fraction decreased over time.	Mouse model of acute myocardial infarction
Human CDC	miR-210	Mitigation of adverse remodeling and improvement of angiogenesis in pig models with myocardium infarct.	In vivo
	miR-146a	Inhibition of apoptosis and stimulation of the proliferation of cardiomyocytes and angiogenesis in vitro. Improvement of heart function in mouse myocardial infarction model.	In vitro, using human umbilical-derived cells, and in vivo mouse model
Human CPC	miR-132	Inhibition of cardiomyocyte apoptosis and improvement of cardiac function after myocardial infarction.	In vitro and in vivo, using rat models
Human ADSC	miR-126	Increased angiogenesis of endothelial cells. Exosomes from ADSC of obese subjects present a reduced load of miR-126 and a low pro-angiogenic capacity.	In vitro, using human umbilical-derived endothelial cells
	miR-31	Increase the migration and tube formation of human umbilical vein endothelial cells.	

MSC—mesenchymal stem cells; ESC—embryonic stem cells; CPC—cardiac progenitor cells; CDC—cardiosphere-derived cells; ADSCadipocyte-derived stem cells; LAD—left anterior descending coronary artery.

## Data Availability

Not applicable.

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
