# Peer review of "Charting the Path: Navigating Embryonic Development to Potentially Safeguard against Congenital Heart Defects"

_jpm, 2023, doi:10.3390/jpm13081263_

Round 1
Reviewer 1 Report
This review describes congenital heart disease from the perspective of embryonic development, its normal development and pathogenesis. This is an excellent summary and evaluation of a thorough survey of the current situation, and it is a paper worthy of publication.
Author Response
Reviewer’s comments: “This review describes congenital heart disease from the perspective of embryonic development, its normal development and pathogenesis. This is an excellent summary and evaluation of a thorough survey of the current situation, and it is a paper worthy of publication.”.
AUTHORS’ ANSWER/MODIFICATIONS:
We are grateful for the warm appreciation of our manuscript. Please note that two additional tables have been added to the manuscript and some light modifications were introduced in the text to revise the English and correct typos, and to clarify some parts in manuscript to address the suggestions made by the Reviewers #2 and #3. The changes/additions made in the text are highlighted in yellow in the revised version of the manuscript. Please note also, that the flow of the main text has not been altered, although the figure has been relocated and new tables were added.
Please see also the attachment.

Reviewer 2 Report
The Authors submitted an interesting review article highlighting key molecular and cellular processes involved in heart development. Subsequently, it explores the potential for future therapeutic strategies targeting the early embryonic stages to prevent CHD by delivering biomolecules or exosomes to compensate for faulty cardiogenic mechanisms. So they conclude that implementing such non-surgical interventions during early gestation may offer a prophylactic approach to reduce the occurrence and severity of CHD.
The topic is interesting and appealing. The manuscript is well-written and presented. The figure is of good quality. The table is clear, but I suggest adding some more Tables, two at least, summarising the major gene implicated in CHD and their function and one more that presents signaling pathways and exosomes in heart development. A light English grammar revision and a double-check for typos is also warranted.
Author Response
Reviewer’s comments: “The Authors submitted an interesting review article highlighting key molecular and cellular processes involved in heart development. Subsequently, it explores the potential for future therapeutic strategies targeting the early embryonic stages to prevent CHD by delivering biomolecules or exosomes to compensate for faulty cardiogenic mechanisms. So they conclude that implementing such non-surgical interventions during early gestation may offer a prophylactic approach to reduce the occurrence and severity of CHD.
The topic is interesting and appealing. The manuscript is well-written and presented. The figure is of good quality. The table is clear, but I suggest adding some more Tables, two at least, summarising the major gene implicated in CHD and their function and one more that presents signaling pathways and exosomes in heart development. A light English grammar revision and a double-check for typos is also warranted.”.
We thank the Reviewer #2 for finding that our manuscript is interesting, and with relevance for the research field. The changes and additions made to address the reviewer’s comments and suggestions are highlighted in yellow in the revised manuscript. We are grateful for the comments and suggestions made, which helped to complete and improve our manuscript.
The Reviewer’s comments are addressed below:
“…I suggest adding some more Tables, two at least, summarising the major gene implicated in CHD and their function and one more that presents signaling pathways and exosomes in heart development.”.
AUTHORS’ ANSWER/MODIFICATIONS:
We have introduced two additional tables in the text. Table 2 presents some genes frequently mutated and associated to congenital heart disease, described the function of these genes in relation to heart development and the cardiac phenotypes associated to their mutation. This table includes signalling pathways activators and mediators.
Table 3 presents some human cell sources with identified active micro-RNA and their effect on cardiovascular system.
We have also revised the English grammar and corrected typos, and introduced slight alterations to improve the text and reading.
Please see also the attachment.

Reviewer 3 Report
The manuscript by Bragança and co-authors is a comprehensive review of the current knowledge on Congenital heart diseases (CHD). The authors described the molecular mechanisms of CHD and the current therapeutic approaches. The review is well-organized and the mechanisms are deeply described.
I have only a minor suggestion.
The authors have to introduce a new Figure describing the complex secretome described in paragraph 3 to facilitate readers.
Author Response
Reviewer’s comments: “The manuscript by Bragança and co-authors is a comprehensive review of the current knowledge on Congenital heart diseases (CHD). The authors described the molecular mechanisms of CHD and the current therapeutic approaches. The review is well-organized and the mechanisms are deeply described.
I have only a minor suggestion.
The authors have to introduce a new Figure describing the complex secretome described in paragraph 3 to facilitate readers.”.
We thank the Reviewer #3 for finding that our manuscript is comprehensive, up to date, well-organized. The changes and additions made to address all reviewer’s comments and suggestions are highlighted in yellow in the revised manuscript. We are grateful for the comments and suggestions made, which helped to improve our manuscript.
The Reviewer’s comments are addressed below:
“The authors have to introduce a new Figure describing the complex secretome described in paragraph 3 to facilitate readers.”.
AUTHORS’ ANSWER/MODIFICATIONS:
Our original idea was to prepare a figure with the main steps of heart development and another with the signalling pathways, as suggested by the Reviewer. However, after reflection, we thought that combining the pathways and mediators in the figure presenting cardiac developmental steps at the times when they are relevant would be more elucidating. The result was the figure presented in the manuscript, with the panel 1A, depicting the development and the panel 1B presenting the signalling pathways/mediators.
On the other hand, as mentioned by the reviewer, the section 3.1 is complex and did not redirect the reader to the Figure 1B, which greatly elucidates the text in our opinion. We have now introduced in the text a reference to the Figure 1B (lane 283), and altered slightly the figure legend corresponding to the panel B. We believe that these small additions will direct the reader to the figure and clarify the text, thus overcoming the need for a novel figure with the description of the secretome, which we think would be presenting information redundant with those in Figure 1B.
Please see also the attachment.
